# Applications of Deep Mutational Scanning in Virology

**DOI:** 10.3390/v13061020

**Published:** 2021-05-28

**Authors:** Thomas D. Burton, Nicholas S. Eyre

**Affiliations:** College of Medicine and Public Health, Flinders University, Bedford Park, SA 5042, Australia; burt0181@flinders.edu.au

**Keywords:** deep mutational scanning, virology, virus, hepatitis, Zika, Dengue, Influenza, SARS-CoV-2

## Abstract

Several recently developed high-throughput techniques have changed the field of molecular virology. For example, proteomics studies reveal complete interactomes of a viral protein, genome-wide CRISPR knockout and activation screens probe the importance of every single human gene in aiding or fighting a virus, and ChIP-seq experiments reveal genome-wide epigenetic changes in response to infection. Deep mutational scanning is a relatively novel form of protein science which allows the in-depth functional analysis of every nucleotide within a viral gene or genome, revealing regions of importance, flexibility, and mutational potential. In this review, we discuss the application of this technique to RNA viruses including members of the Flaviviridae family, Influenza A Virus and Severe Acute Respiratory Syndrome Coronavirus 2. We also briefly discuss the reverse genetics systems which allow for analysis of viral replication cycles, next-generation sequencing technologies and the bioinformatics tools that facilitate this research.

## 1. Introduction

In its essence, deep sequencing is a tool that allows the sequencing of a genomic region multiple times. Deep mutational scanning (DMS) is a technique that utilises deep sequencing technology in combination with a library of mutant genes or genomes produced by random mutagenesis to probe the functional effects of mutations at every single nucleotide position within a gene or genome, linking genotype to phenotype in a single high-throughput experiment. This technique has been applied to various proteins to reveal residue-specific information regarding many aspects of protein biology. For example, under certain conditions, replacement of a yeast gene with a mutant library of a human orthologue can allow for determination of mutants which impact growth, and may be linked to human disease [1,2]. Unbiased selection and identification of mutants with improved properties or activities for specific requirements, such as increased solubility, have furthered the field of protein engineering [3]. Numerous other applications have also been pursued through DMS, including construction of complete functional activity landscapes of genes [4,5], establishment of quantitative evolutionary models [6,7], and contributions to structural biology [8,9]. In this review, we will predominantly focus on how DMS has been applied to the study of viral replicative fitness and immune evasion in the context of infectious virus replication cycles.

First, we should consider how residues and genomic regions of importance have been analysed by point mutagenesis in the past. Many studies have utilised alanine substitution to reveal residues or regions of importance in a viral protein. Commonly, residues for mutation are selected by analysis of protein structure, conservation with other isolates or strains, or amino acid biophysical properties [10,11]. Alternatively, large regions of interest may be interrogated by alanine scanning mutagenesis as a less targeted approach [12,13]. Alanine mutagenesis is the preferred substitution, as alanine features an inert, non-bulky methyl functional group, and does not alter main-chain conformation [14]. In these studies, mutants of interest are typically loss-of-function mutants, as this is indicative of the absense of a functionally important residue. Experiments attempting to generate adaptive or gain-of-function mutations often rely on serial passages of a virus in cell culture or animal models [15,16,17], relying on an error-prone viral polymerase to generate mutants. A limiting factor is that combinations of mutations that may be required to enhance viral fitness in a given host may not be realised within the system. This method can also be time consuming as multiple rounds of infection are required. In regard to analysis of the impact of amino acid substitutions on viral protein function or viral replicative fitness, DMS can allow the substitution of a given residue with all possible amino acids, increasing the probability of identifying gain-of-function mutations that otherwise require more than one nucleotide substitution.

DMS studies in virology are generally performed using a three-step approach. First, a mutant library of a gene or genome of interest is prepared, ideally with genetic variants encoding every single possible residue change within the sequence of interest. Second, a selective pressure is applied to the library, enabling the enrichment of mutations that encode a selective advantage and the removal of deleterious mutations. Finally, the frequency of mutations within the library is quantified via next-generation sequencing (NGS) and compared before and after the application of the selective pressure. In studies of viral replicative fitness, a mutant with a cost to fitness will be selected against, while an enhancing mutant, for example, an antiviral escape mutant, will become enriched. Selective pressures, such as drug/antibody presence, stimulation of antiviral innate immunity, growth in cell types of different species, and binding potential to a host receptor have been applied to studies of many viral genes. Analysis of variants through DMS has enabled evolutionary studies, escape mutant predictions, attenuated vaccine construction and other applications. In this review, we will discuss various DMS strategies and how they have been applied in virology, with a focus on the *Flaviviridae* family of viruses, Influenza Virus A (IAV) and Severe Acute Respiratory Syndrome Coronavirus 2 (SARS-CoV-2). DMS studies of RNA viruses have been made possible through the development of reverse genetics systems, or easily modified infectious clones, and advances in sequencing technologies, which we will discuss herein.

### 1.1. Influenza A

As members of the *Orthomyxoviridae* family, IAVs are negative-sense single-stranded RNA (-ssRNA) viruses [18]. Influenza strains vary from seasonal variants which cause a significant health care burden, to pandemics such as the Spanish Flu which killed an estimated 50 million people from 1918 to 1919 [19]. Influenza circulates not only in the human population, but in species such as birds, pigs and cats [20]. Two major obstacles towards effective Influenza protection and treatment are antigenic shift and antigenic drift, which are mechanisms which, through mutation or gene reassortment, can functionally change Influenza biology. Antigenic drift, which also applies to other viruses such as *Flaviviridae* and *Coronaviridae* family members, is the accumulation of point mutations due to the error-prone virally encoded RNA-dependent RNA polymerase used to generate new copies of the genome during replication [21]. Antigenic shift allows the formation of new Influenza subtypes through reassortment of surface antigens haemagglutinin (HA) or neuraminidase (NA) and is facilitated by the segmented genome of Influenza. Antigenic shift occurs when an animal is infected with two or more different strains of Influenza virus, enabling reassortment of HA and NA viral RNA segments and leading to a novel Influenza strain to which the population is completely naïve [21]. This resulted in the first pandemic of the 21^st^ Century: swine-origin IAV [22]. The prediction of mutant strains of IAV that are likely to arise is therefore imperative for prophylaxis and control, and the predictive power of DMS can be utilised for the identification of animal viruses with zoonotic potential, or to predict escape mutants that are insensitive to otherwise effective adaptive immune responses or antiviral therapies.

### 1.2. Flaviviridae

The *Flaviviridae* family of RNA viruses comprises multiple positive-sense single-stranded RNA (+ssRNA) enveloped viruses, including hepatitis C virus (HCV), Dengue virus (DENV) and Zika virus (ZIKV) [23]. These viruses have enormous impact around the globe, with DENV alone infecting approximately 390 million people each year [24]. For most *F**laviviridae* (and viruses in general), no therapeutics or vaccines exist, with exceptions for vaccines being yellow fever virus [25] and Japanese encephalitis virus [26]. In addition to furthering our understanding of basic *Flaviviridae* biology, DMS has enabled the construction of a vaccine candidate for ZIKV, which will be discussed later.

### 1.3. SARS-CoV-2

The *Coronaviridae* family of positive-sense single-stranded RNA (+ssRNA) viruses features several clinically relevant viruses including Middle East respiratory syndrome-related coronavirus (MERS-CoV), Severe Acute Respiratory Syndrome Coronavirus (SARS-CoV), and the current pandemic SARS-CoV-2, all of which have emerged in the past two decades [27], as well as common cold-causing viruses such as HCoV-OC43 [28]. There is currently a major global research effort to develop therapeutics and vaccines against SARS-CoV-2 which ideally target regions of the virus that are incapable of mutational escape, which would otherwise reduce the effectiveness of the treatment. DMS has been employed to determine how mutants will affect SARS-CoV-2 neutralisation.

## 2. Reverse Genetics Systems to Study RNA Viruses

Due to the absence of DNA in the lifecycle of RNA viruses, the construction of cDNA clones of infectious RNA viruses has become an incredibly important tool in studying their lifecycle. Generally, a reverse genetics system is comprised of genomic viral RNA that is reverse-transcribed into cDNA and then cloned into a plasmid, allowing for stable propagation of a virus genome within bacteria or yeast. The plasmid can then be easily manipulated by standard molecular methods, with the introduction of mutations and tags, or the removal of segments of the genome [29].

There are multiple key requirements for an infectious (+)RNA virus cDNA clone. A common approach involves incorporation of a DNA-dependent RNA polymerase promoter at the 5′ end of a cloned viral genome to enable in vitro transcription of infectious viral RNA via a corresponding RNA polymerase, with or without a type I 5′ cap structure, if required. This is commonly achieved using bacteriophage promoters/polymerases such as those of T7 and SP6. Produced RNA can then be purified and transfected into cells to initiate the viral replication cycle. A constitutive promoter such as a human cytomegalovirus [HCMV] promoter can also be utilised, with viral RNA produced by host Polymerase II after direct transfection of a full length cDNA clone into cells. As it is necessary to produce viral RNA with precise ends, a self-cleaving hepatitis delta or hammerhead ribozyme or a T7 terminator sequence may be added to the 3′ end of the viral genome to enable generation of authentic 3′ ends [30,31]. Alternatively, a unique restriction endonuclease site can be inserted at the 3′ end for plasmid linearization [32]. Some of the limitations of these plasmid systems are instability in bacteria, due to the presence of cryptic bacterial promoters and other factors leading to recombination during growth, and poor plasmid yields [33]. As the preparation of mutant libraries involving plasmid clones of viral cDNA often requires the pooling of a large number of bacterial colonies, it is important in DMS studies to have a plasmid with both minimal recombination, to ensure that recombination of the plasmid in bacteria does not affect cell culture experiments/analysis, as well as high transformation efficiency to ensure that a library of sufficient mutational diversity can be prepared. For these reasons, several bacterium-free approaches such as circular polymerase extension reaction (CPER) [34] and yeast artificial chromosome (YAC) approaches, including transformation-associated recombination (TAR) cloning in *Saccharomyces cerevisiae* [35], have also been employed to enable efficient propagation and manipulation of (+)RNA virus cDNA clones. Similarly, bacterial artificial chromosome (BAC) systems and modified plasmid DNA clones with features designed to minimise viral cDNA recombination and toxicity in bacteria have also been applied to various reverse genetics systems for (+)RNA viruses [31]. The reliability and ease of manipulation of viral cDNA using these systems are important determinants of the success of DMS experiments.

Many significant hurdles were encountered during the construction of an Influenza reverse genetics system. The Influenza genome comprises eight negative-sense ssRNA segments [18]. Due to the negative-sense genome, the viral RNA is not sufficient for initiation of viral replication, with the presence of multiple viral proteins being required to begin the viral lifecycle [18]. The eight vRNAs must colocalise with this protein machinery within the nucleus for initiation of infection [18]. Many reverse genetics systems have been developed to overcome these difficulties, with the two main categories comprising helper virus-dependent and -independent systems. Please see the review by Neumann et al. for a more in-depth analysis of the history of Influenza reverse genetics systems [36]; and for detailed documentation of these systems, please see the review by Engelhardt et al. [37].

Reverse genetics systems exist for many Flaviviridae species. Unlike Influenza A, the construction of infectious clones for *Flaviviridae* was, at the time, hindered more by the lack of available technologies such as high-fidelity PCR, instead of the fundamental biology of the virus. Construction of these systems is often considered as relatively straightforward although, as detailed above, challenges in genome construction due to repetitive elements, toxicity in *E. coli* and associated instability and recombination in *E. coli* are well-documented. For a history of Flavivirus reverse genetics systems, please see the review by Aubry et al. [31]. 

A reverse genetics system has recently been developed for SARS-CoV-2 [35]. However, as no DMS experiments have been performed to date using a reverse genetics system for the virus, we will not discuss this further. However, we highlight this as a potential area for future DMS studies.

Reverse genetics systems have traditionally been used in low-throughput mutational studies, to analyse the effect of single point mutations. Coupling reverse genetics with deep sequencing and random mutagenesis has allowed for high-throughput mutational studies, with analysis of hundreds of thousands of mutants being made possible in a single experiment. In the following sections, we discuss the technologies which have been essential in enabling DMS studies.

## 3. Next-Generation Sequencing

Advances in next-generation sequencing (NGS) have been crucial to the development of DMS as a tool in the field of molecular virology. The first generation of sequencing consisted mainly of Sanger sequencing and the Maxam and Gilbert technique. The second generation of sequencing introduced mass parallelisation of reactions. The current generation of sequencers enable real-time, single molecule sequencing [38]. We will discuss three commonly used platforms in DMS studies; Pacific Biosciences (PacBio) Single Molecule Real-Time (SMRT) sequencing, Oxford Nanopore Technologies (ONT) real-time sequencing, and the Illumina short-read sequencing-by-synthesis technology, which is part of the second generation of sequencing. [39]. For further in-depth detailing of the latest advances in NGS, please see the review by Goodwin et al. [40].

### 3.1. Single-Molecule Long-Read Sequencing

Produced by Pacific Biosciences, PacBio SMRT technology (henceforth referred to as SMRT) utilises a sequencing by synthesis approach. Initially, the DNA is fragmented into pieces several kilobases in length and the addition of hairpin adaptors to the DNA results in the formation of a circular SMRTbell DNA conformation. Next, the circular DNA is introduced to a flow cell lined with picolitre wells with a transparent bottom (a zero-mode waveguide) and a DNA polymerase enzyme fixed to the bottom of the well. The polymerase then incorporates a fluorescently tagged nucleotide into the elongating DNA strand, and the fluorescent signal emitted by the individual nucleotide being incorporated is recorded by a camera. Finally, each fluorophore is cleaved by the polymerase and diffuses before the next read occurs. A major advantage of this technology is that the circular SMRTbell DNA conformation allows for many rounds of sequencing of a single DNA fragment, producing an accurate circular consensus sequence [41] (Figure 1).

The Oxford Nanopore system (henceforth referred to as ONT) sequences DNA in a unique manner. DNA is fragmented into pieces of several kilobases in length, then a motor protein and a hairpin adaptor are added to either side of the DNA. A leader sequence directs the DNA to a pore embedded in an electrically resistant membrane, and the motor protein allows ssDNA to be pulled through the aperture of the pore. Simultaneously, an electric current passes through the pore protein. As the ssDNA passes through the pore, a characteristic disruption of the electrical current occurs, dependent on the multiple bases present in the pore. By analysis of this disruption, a DNA sequence can be identified by its unique ‘k-mer’, which can be translated into a sequence (for example, AAGT will have a distinct disruption compared to AGAT). The hairpin adaptor allows for bidirectional sequencing of the DNA fragment, and the read from the forward and reverse strands can be used to generate a consensus sequence [42] (Figure 2).

### 3.2. Illumina Sequencing by Synthesis

Used in Illumina sequencing instruments, DNA molecules are sheared to ≤300 base pair fragments then ligated to adapter sequences which allow for hybridisation to complementary oligonucleotides present in nanowells across a patterned flow cell. The DNA fragments are amplified via bridge amplification, resulting in clonal clusters of DNA. Subsequent addition and imaging of fluorophore-labelled terminator nucleotides allows for highly parallel sequencing [43] (Figure 3).

### 3.3. Next-Generation Sequencing Technologies: A Comparison

The main limitation of Illumina short-read technology in DMS studies is the short-read length. Viral segments for analysis are often much longer than the read length of the instrument. If two distant mutants are present in a single viral genome, and a specific phenotype is observed, it is difficult to determine without further experimentation if they are acting in a pairwise manner or if a single mutation is wholly responsible for the phenotype. Thus, pairwise epistatic mutations are not always resolvable using short-read technology. A partial solution to this problem is to create multiple adjacent mutant libraries of short lengths, ‘tiling’ across a gene, to restrict epistatic mutations to within a readable window [44]. Alternatively, subassembly can be utilised. In this approach, tagging each template molecule with a unique DNA barcode or ‘unique molecular identifier’ (UMI) allows for the grouping and analysis of multiple short reads on the basis of their original template molecule [45]. Resolution of distant mutants is only possible using third generation long-read sequencing strategies, offered by both SMRT and ONT.

Of the long-read sequencing platforms, SMRT is far more common than ONT in DMS studies, likely attributed to its higher accuracy. The circular SMRTbell DNA conformation allows for increased sequencing depth (how often a base is sequenced on average) to form a consensus sequence, as the SMRTbell can be repeatedly sequenced. Errors in PacBio sequencing are distributed randomly, and therefore the accuracy increases with increased reads of a single SMRTbell molecule [46]. Oxford Nanopore sequencing utilises dsDNA fragments, but does not allow continuous sequencing. A consensus sequence can be formed by sequencing one strand of dsDNA, followed immediately by the complementary strand, referred to as 1D^2^ sequencing, though the accuracy is somewhat limited compared to PacBio and Illumina strategies [46]. Interestingly, a recent study has reported a high-throughput amplicon sequencing approach that combines UMIs with PacBio or ONT to enable generation of high-accuracy single-molecule consensus sequences for large DNA regions [47]. This and similar approaches will help to overcome compromises in accuracy that have previously been associated with the above long-read sequencing strategies. To date, however, Illumina sequencing remains the most commonly used platform in DMS studies due to its cost-effectiveness and high levels of accuracy.

## 4. Deep Mutational Scanning: Library Construction

As mentioned previously, the first step in a DMS experiment is the construction of a mutant library. Multiple approaches have been developed and applied to mutant library generation, which we will briefly describe below.

The simplest and most cost-effective method is error-prone PCR. In this method, one or more polymerases are employed to exponentially amplify a region of DNA, with initial template amount and cycle number varied to optimise mutation rate [48]. *Taq* DNA polymerase, a popular error-prone polymerase, has a reaction buffer-dependent mutation rate of roughly 8 × 10^−6^ errors/nucleotide [49]. However, the use of Taq polymerase alone in library construction is limited by the mutants generated being dominated by AT → GC transitions and AT → TA transversions, at approximately 2-4x the mutation rate of G and C residues. A polymerase named Mutazyme DNA polymerase has a 2-4x stronger preference for GC → AT transitions and GC → TA transversions [50]. In combination, these enzymes can be used to produce a relatively unbiased mutant library with a somewhat controllable mutation rate. A major drawback of this method is that not all amino acid residues are accessible with single-nucleotide polymorphisms [51]. Additionally, this technique can often result in multiple mutations present in a single DNA fragment that can confound results, especially when using short-read sequencing strategies as mentioned earlier.

First described in virology studies by the Bloom lab in an Influenza nucleoprotein (NP) study, synthetic oligonucleotides were designed to contain a randomised triplet for each codon present in the gene, with 16 leading and lagging nucleotides that anneal specifically to the NP gene, as well as the reverse complement of these oligonucleotides. Using a series of joining PCR steps and restriction enzyme cloning, a product pool containing all possible amino acid mutations of NP was cloned into an Influenza reverse genetics system [6]. When designing oligonucleotides with randomised triplets, codon usage should be considered. NNK degeneracy (N: Ade/Cyt/Gua/Thy, K: Gua/Thy) encodes all amino acids, while NDT (N: Ade/Cyt/Gua/Thy, D: Ade/Gua/Thy, T: Thy) and DBK (D: Ade/Gua/Thy, B: Cyt/Gua/Thy, K: Gua/Thy) each encode 12 amino acids, feature no stop codons and exhibit all major biophysical types, while potentially decreasing workload. However, with decreased coverage, the amount of interesting variants will also be reduced [52]. This method is more costly compared to the error-prone PCR method, but can be used to generate libraries with complete mutational coverage.

A synthetic approach to mutant library generation is also available. Gene synthesis begins with oligonucleotide construction. These are designed such that adjacent oligonucleotides in the final product contain overlapping sequences. The overlap results in a DNA duplex which, through an assembly reaction with DNA polymerase, results in the construction of the gene of interest [53]. Multiple methods allow for controlled or randomised insertion of mutants into a gene. For example, the “Spiked Genes” method uses oligonucleotides spiked, or interspersed, with mutants to create a pooled mutant fragment [54]. Synthetic construction of multiple types of mutant libraries is possible through commercially available gene synthesis services. Available mutant library types include controlled randomised libraries, scanning alanine mutagenesis libraries, scanning codon mutagenesis libraries and others.

An additional type of mutant library discussed in this review is the transposon insertion library. This is facilitated by a transposase protein, such as Tn5, Tn7 or MuA that recognizes the ends of a transposon, or a mobile DNA element, and forms a protein-DNA complex termed the ‘transpososome’. This complex catalyses cleavage of target DNA at a random site (although evidence of insertion bias exists [55]), and joining reactions allow for the introduction of the transposon into a DNA template. For many applications, the transposon is engineered to feature an antibiotic resistance gene, allowing for selection of genetic elements with successful transposon integration. In these systems the transposon also features two identical restriction enzyme sites on each end of the DNA element. Upon purification, the now-unique plasmids are digested with the required restriction enzyme. Gel electrophoresis allows for the removal of the majority of the introduced DNA element by size separation, with a small insertion remaining after ligation of the plasmid backbone [56,57]. High-throughput random mutagenesis of a cloned viral genome paired with deep scanning allows for incredibly powerful studies which can probe regions of genomic flexibility and functionality.

## 5. Standard Deep Mutational Scanning Experiment Methodology

DMS experiments often follow a similar methodology. Using a reverse genetics system, a mutant virus library is generated via a method such as randomised mutagenesis, controlled site-saturation mutagenesis, or transposon mutagenesis. This library is often initially amplified using bacteria, with a higher number of uniquely transformed plasmids translating to increased mutational diversity. This library is transfected into cells, allowing for the propagation of RNA replication-competent viral genomes, as well as production of viral particles. Infection of naïve cells with virus-containing cell culture supernatants allows for propagation of genomes which are capable of both RNA replication and infectious particle production. Variant analysis by NGS is then key to identifying specific variants which are present at higher or lower levels at certain stages of the viral lifecycle in comparison to the initial input (Figure 4).

While not a major focus in this review, it is important to note that DMS studies of individually expressed proteins have been highly informative, particularly in the context of the interactions of viral glycoproteins with host receptors or neutralising antibodies. For example, angiotensin-converting enzyme 2 (ACE2), the host receptor for SARS-CoV-2 [58], has been analysed for mutants which enhance binding to the spike glycoprotein, to explore the use of an engineered soluble ACE2 as a potential therapeutic [59,60]. The binding of computer-generated miniprotein inhibitors [61] and ACE2 decoys [62] to the spike glycoprotein has also been effectively optimised using DMS. A similar strategy was used to optimise affinity and specificity of a computationally designed protein that targets H1N1 Influenza haemagglutinin [63,64].

## 6. Bioinformatic Tools for Deep Mutational Scanning Data Analysis and Visualisation

Multiple software packages are available which facilitate variant analysis. ‘dms_tools’ [65] and ‘Enrich2’ [66] calculate amino acid preferences under a selective pressure. ‘dms-view’ allows straightforward visualisation of DMS data in the context of a protein structure [67]. ‘MaveDB’ (multiplex assays of variant effect database) provides a tool for sharing DMS data analyses.

## 7. Deep Mutational Scanning Experiments

In this section, we will briefly describe several past applications of DMS in virology, focussing on *Flaviviridae* family viruses, IAV and SARS-CoV-2. Transposon mutagenesis has been applied to several viruses to identify genomic regions tolerant to small insertions. An insertion with minimal impact on fitness indicates genomic flexibility, or high mutability, at a specific area, which may allow for adaptation of the virus to a new environment. Disruption of function from an insertion may also reveal a region of functionality within a gene or protein. The identification of a region of insertional tolerance is often exploited by incorporation of epitope tags or reporter genes for further research. DMS projects focussing on single-nucleotide polymorphisms and codon mutagenesis have focussed on several biological aspects, including identifying determinants of viral tropism, epistatic interactions, drug and antibody escape mutants, and residues critical to several biological functions. It is important to note that the following summary is not a comprehensive list of all DMS scanning projects. DMS has also been applied to viruses including Norovirus [68], Venezuelan equine encephalitis virus [69], Measles virus [70], Paramyxoviruses [71], foot-and-mouth disease virus [72], Epstein–Barr virus [73] and Human immunodeficiency virus [74,75,76,77,78,79], and as mentioned previously, we focus mainly on studies of viral replicative fitness and immune evasion.

### 7.1. Hepatitis C Virus (HCV)

#### 7.1.1. Transposon Analysis

In a comprehensive early study by Remenyi et al., the Mu-transposon system was used to insert 15 nucleotide (nt) sequences (of which 10 nt are transposon derived and 5 nt are duplicated target sites) into an HCV plasmid based on the chimeric sequence of genotype 2a J6 and JFH1 viruses [80]. This mutant library was propagated in human hepatoma Huh-7.5.1 cells, then passaged onto naïve Huh-7.5.1 cells to separately analyse genomes capable of viral RNA replication and infectious particle production. Regions tolerant to an insertion illustrate flexibility at the genomic level, and potentially highlight a region non-essential for in vitro growth. P7 and non-structural protein 2 (NS2), each coordinators of virus assembly [81], and envelope protein 2 (E2) were identified as potential areas for insertion of a tag or small peptide with minimal costs to replicative fitness due to the regions of high genomic flexibility uncovered. The impact of NS4B transposon insertions on infectious particle production helped to reveal a previously unknown functional region of NS4B which was suggested to play a role in the viral lifecycle post-RNA replication.

In a subsequent study by the same group, a similar methodology was used to assess insertions that confer sensitivity to the antiviral cytokine Interferon alpha (IFN-α) [82]. After two rounds of passage in Huh-7.5.1 cells in the presence or absence of IFN-α, it was observed that mutants conferring sensitivity to IFN-α were clustered in p7, the 3′ untranslated region (UTR) and non-structural protein 5A (NS5A), which is important for viral replication, infectious virus particle production and modulation of host cell signalling [83]. Eight IFN-α sensitive p7 insertion mutants were constructed for validation, and revealed the role of p7 in immune evasion. An interferon-stimulated gene (ISG) cDNA expression library screen demonstrated that 13 of these ISGs were particularly antiviral against viruses with mutant p7 in comparison to wild-type p7. Amongst these ISGs, IFI6-16 was identified as a major target of p7-mediated immune evasion. Coimmunoprecipitation was then used to show a direct interaction between p7 and IFI6-16, which was previously unknown. Each of these DMS studies furthered understanding of the roles of HCV proteins in infection.

#### 7.1.2. Variant Analysis

In an important application of DMS towards understanding emergence of viral resistance to antiviral drugs Qi et al. applied high-throughput variant analysis towards understanding resistance to Daclatasvir [84]; a potent inhibitor of NS5A [85]. An NS5A mutant library of the J6/JFH1 strain (genotype 2a) was prepared and passaged in Huh7.5.1 cells in the presence or absence of daclatasvir over 2 passages, before analysis by NGS. Drug selection resulted in selection of drug-resistant mutants at NS5A residues 28, 31, 38, 92 and 93, and increased drug resistance of some mutants at four additional residues. These escape mutants can be used to identify daclatasvir-resistant strains of HCV [86]. In support of this methodology, many of these resistance mutations have been identified in numerous in vitro and clinical studies of HCV resistance to Daclatasvir and related inhibitors [87].

### 7.2. Zika Virus (ZIKV)

#### 7.2.1. Transposon Analysis

In the first published study involving high-throughput mutational analysis of ZIKV a transposon insertion mutant library was generated for the ZIKV MR-766 strain [88]. Human embryonic kidney 293T (HEK 293T) cells were used to propagate the virus, then naïve African green monkey kidney Vero cells were infected with viral supernatants over two rounds. The first pool of selected viral RNA, reflecting viral RNA replication requirements in HEK 293T cells, showed high levels of flexibility in the structural proteins Envelope (E), precursor membrane (prM) and capsid (C), and also non-structural protein 1 (NS1), which is essential for ER remodelling and RNA replication [89]. The third pool, consisting of infectious particle production-competent genomes, displayed most flexibility in the structural proteins. Some insertions present in NS1 rendered the virus incapable of infectious particle production while enhancing viral RNA replication capacity. In this context, an insertion after amino acid 174 resulted in >650-fold enrichment in the initial HEK 293T cell viral RNA population compared to the input RNA library yet was not present in the Vero cell populations. Together, this study revealed regions of genetic rigidity in the ZIKV genome across multiple lifecycle stages.

#### 7.2.2. Variant Analysis

##### Envelope

Several mutational analyses of the ZIKV Envelope (E) protein have been performed. An envelope mutant library for the MR766 ZIKV strain was constructed by Sourisseau et al. to examine how mutants affect viral neutralisation by monoclonal antibodies [90]. The MR-766 strain was first tested for the effects of mutations on viral fitness, with a focus on infectious particle production in Vero cells. After validation, mutational antigenic profiling was performed, whereby the selective pressure of a neutralising antibody results in the enrichment of escape mutants present in the mutant library. Two antibodies were used in this screen; ZKA64, which binds domain III of recombinant E protein, and ZKA185 which neutralises ZIKV without interaction with E protein, or any confirmed epitope [91]. Domain III of E protein is responsible for binding the cellular receptor, and is a common target of potent ZIKV-neutralising antibodies [92]. The mutant ZIKV library was subjected to these antibodies individually. ZKA64 and ZKA185 treatment resulted in strong selection of several mutations that corresponded to sites within E, with mutants selected by ZKA64 present on envelope domain III, and mutants selected by ZKA185 present on envelope domain II. Antibody escape resistance mutations A333T and T335E for ZKA64, and D67A and K118R for ZKA185, were confirmed in follow-up neutralisation assays, illustrating the usefulness and accuracy of mutational antigenic profiling for escape mutant predictions, and demonstrating that antibody escape mutants can appear outside the receptor binding domain.

The impact of Envelope on tropism using ZIKV strain PRVABC59 has also been examined [93]. This study analysed mutant library growth in three cell types: C6/36 cells, derived from *Aedes albopictus* larvae, A549 adenocarcinomic human epithelial cells and hCMEC/D3 cells, a human blood–brain barrier endothelial cell line. NGS analysis revealed that a mutation at N154, an N-linked glycosylation site on the ZIKV envelope protein, resulted in a significantly higher fitness of ZIKV in mosquito cells, but not in human cells. Additionally, ablation of glycosylation through mutations surrounding the N154 position enhanced viral entry into C6/36 cells, resulting in increased levels of viral replication. DC-SIGN is a known viral entry factor for ZIKV that is not expressed in 293T cells such that overexpression of DC-SIGN in 293T cells enhances ZIKV infection [94]. A comparison between wild-type ZIKV and ZIKV glycosylation mutants showed that, with overexpression of DC-SIGN, infection of 293T cells by the glycosylation mutant ZIKV was significantly diminished compared to wild-type ZIKV, demonstrating the requirement of N-linked glycosylation of E in viral entry into mammalian cells, despite the resultant loss of fitness in mosquito cells.

In a similar study into cell tropism, a mutant library for the C-terminal of the E protein of ZIKV (PRVABC59) was prepared and grown in Vero and C6/36 cells for 8 and 13 days, respectively, to identify host-adaptive substitutions [95]. In C6/36 cells, mutants K316Q and S461G were preferentially selected. The K316Q/S461G virus replicated less efficiently in a selection of human cell lines, with lower levels of cytotoxicity compared to wild type, while no change was seen in C6/36 and Aag2 mosquito cell lines. This combinatorial mutant was determined to decrease the thermal stability of E, and as mosquito cell lines are typically grown at a lower temperature (28 °C) than human cell lines, the effect was not observed in the mosquito cell line. ZIKV infection can result in severe developmental defects in the brain [96]. The K316Q/S461G virus was used to infect induced-pluripotent stem cell-derived human brain organoids. Development of these organoids infected with the K316Q/S461G virus greatly decreased growth retardation in comparison to wild-type ZIKV. Infection of interferon alpha receptor knockout mice showed attenuated infection by the K316Q/S461G virus, and protection against a later wild-type ZIKV infection, indicating its potential as an attenuated vaccine candidate.

### 7.3. Dengue Virus (DENV)

#### Transposon Analysis

To identify regions of genetic flexibility in DENV, our group applied the Mu transposase system to generate a library of mutants containing single random 15 nt insertions in a serotype 2 Dengue virus (DENV-2) genome (strain 16681) [97]. Analysing both viral RNA replication and infectious particle production in Huh7.5 cells, non-structural protein NS1, required for RNA replication [98] and virus assembly [10], as well as structural protein capsid (C) demonstrated highest tolerance to the insertions. Lowest tolerance of insertions was seen in structural protein prM and non-structural protein 2A (NS2A), which is essential for virus assembly [99]. Across the genome, flexibility peaked at the C termini for C, E, NS1, NS2B and NS4B, and at the N termini of NS3 and NS4B. These experiments allowed for the tagging of NS1 with epitope tags and reporter proteins including FLAG, APEX2 and NLuc, with minimal loss of viral fitness in Huh7.5 cells.

A similar transposon mutagenesis study of DENV-2 (16681) was performed by Perry et al. [100]. This study identified regions of genetic flexibility and enabled identification of well-tolerated insertion sites within non-structural protein 4B (NS4B), which is involved in membrane rearrangements [101], and C, which were utilised to create a DENV-2 clone expressing HA-tagged capsid, incapable of infectious particle production, and a replication-competent infectious DENV-2 clone expressing HA-tagged NS4B. Together, these studies identified regions of genetic flexibility within DENV-2 and exploited this information to generate infectious reporter virus tools.

### 7.4. Influenza A Virus (IAV)

#### 7.4.1. Transposon Analysis

Transposon mutagenesis has also been applied to great effect with the Influenza (H1N1) A/Puerto Rico/8/1934 strain [102]. Mutagenized viral genome segments were transfected into HEK 293T cells, then viruses were passaged onto naïve Madin–Darby canine kidney (MDCK) cells for two rounds. NGS analysis revealed that two genes were tolerant to insertions; haemagglutinin (HA), an attachment factor and membrane fusion protein [103], and non-structural protein 1, responsible for immune evasion [104]. The mutant library was then grown in embryonated chicken eggs, to determine how the library would perform under different environmental pressures. HA features a head and stalk domain [105]. In chicken eggs, insertions in the head of HA were preserved, while mutants present in the stalk domain were not tolerated. Analysis of these HA head insertion mutants showed no loss of fitness and were found to occur at antigenic sites. The authors hypothesised that this flexibility may reflect a strategy in evasion of host immune systems.

#### 7.4.2. Variant Analysis

##### Hemagglutinin and Neuraminidase

A major focus of IAV research is the functional analysis of NA, which aids virus budding, and HA, as antigenic variation of these proteins is responsible for the limited protection that vaccines provide against IAV infection [106]. Using a HA mutant library, Thyagarajan et al. illustrated that the antigenic sites of Influenza (H1N1) A/WSN/1933 HA are most tolerant to mutations, providing an explanation for Influenza’s rapid antigenic evolution [107]. Similar effects were also reported in DMS studies by Wu et al. [108] and Boud et al. [109]. Influenza (H3N2) A/Perth/16/2009 HA has also been subject to analysis by DMS [110]. Interestingly, from these studies, mutational tolerance was demonstrated to be higher in the HA head for H1N1 Influenza strains, while mutational tolerance was higher in the HA stalk of H3N2 [110]. H3 virus DMS data was therefore shown to not be suitable for prediction of H1 virus evolution, while similarly H1 virus DMS data was not useful for prediction of H3 virus evolution.

Continuing research into HA and its immunogenicity, Doud et al. analysed the effects of single HA mutations on broad and narrow antibody neutralisation [111]. Broad neutralising antibodies are capable of neutralising multiple strains of a virus subtype, or neutralising multiple types of Influenza virus, due to their targeting of conserved regions of proteins, while narrow antibodies are strain specific [112]. Wild-type Influenza (H1N1) A/WSN/1933 was shown to be neutralised by broad anti-RBS and anti-stalk antibodies, and narrow anti-head antibodies. Doud et al. prepared HA mutant libraries of a lab-adapted Influenza (H1N1) A/WNS/1933 strain carrying green fluorescent protein in the PB1 segment and demonstrated that narrow anti-head antibodies as well as broad anti-RBS antibodies were highly susceptible to single mutation escape mutants, while broad anti-stalk antibodies were highly resistant to escape mutants.

Expanding on the single-strain studies of HA, multiple H3N2 IAV strains were investigated for strain-specific HA properties in a recent study by Wu et al. [113]. Five major antigenic sites (A-E) exist for H3N2 HA [114]. To investigate antigenic site B, mutant libraries of H3N2 strains A/Hong Kong/1/1968, A/Bangkok/1/1979, A/Beijing/353/1989, A/Moscow/10/1999, A/Brisbane/10/2007, and A/North Dakota/26/2016 were prepared. These libraries featured specific mutants of the HA receptor binding site at residues 156, 158, 159, 190, 193 and 196; each of which are known to affect receptor binding and viral fitness [115,116,117,118]. A local fitness landscape of each mutant for each strain was constructed, by transfection of the libraries into HEK 293T cells, and subsequent passage of the resultant virus onto MDCK-SIAT1 cells. Analysis of the fitness landscape of these strains illustrated that evolutionary constraints at these residues has changed during the natural evolution of H3N2, indicating that deep mutational scanning data is not always suitable for extrapolation across strains.

In another application of DMS to understand the evolution of IAV resistance to antiviral selective pressures, Wu et al. investigated the antiviral therapeutic Oseltamivir [119], which targets NA of various IAV strains [120]. Oseltamivir treatment has resulted in the generation of Influenza (H1N1) H274Y neuraminidase mutants which decrease sensitivity of the enzyme to oseltamivir. Despite fitness costs associated with the H274Y mutant, the mutant has quickly become widespread [121]. Studying the Influenza (H1N1) A/WSN/1933 virus with the H274Y mutant present, Wu et al. prepared a NA mutant library. After successive passages in A549 cells, four mutants were discovered which either fully (R194G, E214D) or partially (L250P, F239Y) restore H247Y A/WSN/1933 virus to wild-type fitness levels without restored susceptibility to oseltamivir. This study illustrates the power of DMS to understand and predict drug-resistant mutant strains and second site mutations that restore fitness of otherwise attenuated drug-resistant virus strains.

##### Polymerase

Another IAV machinery that has been the subject of DMS studies is the viral polymerase. One of the most comprehensive studies of this nature involved analysis of adaptation of the complex from avian to human cells by Soh et al. [122]. The core of the Influenza virus RNA polymerase consists of the polymerase acidic protein (PA) C-terminal domain, polymerase basic protein 1 (PB1) and the N-terminal of polymerase basic protein 2 (PB2) [123]. PB2 is a known virulence determinant [124] and it has been shown that a single amino acid substitution (E627K) allows avian IAV to replicate efficiently in mammalian cells [125]. Bioinformatics analyses reviewing IAV sequence databases have revealed other mutants which have facilitated host adaptation in the past [126,127]. However, these databases are likely incomplete and limited to past outbreaks. To address these ambiguities, Soh et al. constructed a PB2 mutant library to reveal mutants of Influenza (H7N9) A/Green-winged Teal/Ohio/175/1986 PB2 which improve either polymerase activity or viral replication in human A549 cells in comparison to avian embryo CCL141 cells, to predict mutants which may allow expansion of transmission towards mammalian species. Thirty-three previously undescribed adaptive mutants were discovered. These adaptive mutations were clustered in patches, potentially revealing regions of PB2 responsible for host cell interactions.

Ozawa et al. applied a novel DMS approach to investigate avian Influenza polymerase mutant residues of PA, PB1 and PB2, and their effect on host range [128]. A virus possessing vRNA of both avian Influenza (H5N1) A/Muscovy duck/Vietnam/TY93/2007 and a laboratory-adapted Influenza (H1N1) A/WSN/33, including a partial HA deletion mutant featuring GFP was constructed. Growth of the virus was therefore limited to HA-expressing cell lines. Mutant libraries were generated for PA, PB1 and PB2. The mutant virus library was grown in 293-HA cells, and the highest GFP-expressing 293-HA cells were individually sorted by flow cytometry for infection of naïve MDCK-HA cells. Analysis of expression of matrix protein by quantitative reverse transcription PCR was used to identify 90 viruses for further analysis by deep sequencing. After additional testing, 11 mutants were further characterised in lung and bronchial epithelial cells as well as chicken fibroblast cells, for viral kinetics and polymerase activity, and then for viral replication and pathogenesis in mice, identifying multiple mutants which may increase virulence in mammals.

Continuing the molecular interrogation of PA, Wu et al. coupled DMS with in silico mutant stability predictions to identify functional regions of Influenza (H1N1) A/WSN/1933 PA [129]. A PA mutant library of this strain was used to infect A549 cells, allowing identification of both functionally and structurally important residues. In addition, a protein stability predictor, Rosetta, was used to identify residues of structural importance [130]. Pairing these methods of analysis allowed exclusion of structurally important residues from residues that were deemed to be deleterious by DMS, isolating only functional effects. The study identified multiple residues of PA which abolish polymerase activity with no detectable destabilization effect, as well as residues with little impact on polymerase activity, but a substantial decrease in infectious particle production. While most mutational studies focus on conserved regions for protein functionality, which can exclude species- and strain-specific functional residues, this study by Wu et al. illustrated an unbiased means of determining residues that are strictly functionally important, as opposed to structurally important.

##### Assorted Viral Proteins and DMS Applications

Another application of DMS that is gaining attention involves analysis of viral determinants of sensitivity to innate immune responses. As an example of this, Ashenberg et al. explored resistance to the antiviral host factor MxA [131]. IAV nucleoprotein (NP) forms the viral ribonucleoprotein complex with vRNA and the heterotrimeric RNA-dependent RNA polymerase, and acts as an elongation factor [123,132]. Antiviral host factor MxA is known to suppress viral transcription through its interaction with polymerase PB2 subunit and NP [133]. IAV strains have been identified which either block interferon-induced production of MxA, or, through adaptive mutations, directly confer resistance to MxA [134,135]. Ashenberg et al. determined the impact of all mutant NP residues of Influenza (H3N2) A/Aichi/2/1968 on MxA resistance in MDCK-SIAT1 canine kidney cells either wild type or constitutively expressing MxA. Twenty nine mutants impacting MxA resistance were identified via the screen, with further analysis identifying mutants which increase sensitivity, or increase resistance.

Viral resistance and sensitivity to the antiviral effects of interferons (IFNs) is another emerging application of DMS methodology, as exemplified by a recent study by Du et al. [136]. Many interactions between IAV and the host immune system, as well as host evasion strategies, have long been a focus of interest [137]. Du et al. performed a genome-wide mutant screen for Influenza (H1N1) A/WNS/1933 to discover mutants that alter interferon sensitivity. Virus was grown first in HEK 293T cells, then passaged in A549 cells in the presence or absence of type 1 interferon pre- and post-infection, directly prior to analysis of mutant abundance. Twenty six missense mutations that decreased relative fitness under interferon treatment were randomly selected for further analysis. Eight mutants from viral polymerase subunit PB2, matrix protein M1 and non-structural protein 1 were chosen for further investigation, and then combined to create an interferon-sensitive virus. This mutant IAV showed no attenuation in IFN-deficient Vero cells, but significant attenuation in IFN-competent A549 cells compared to the wild-type IAV strain. Additionally, the virus was attenuated in IFN-competent mice and ferrets, but not IFN AR-/- mice, and acted as an attenuated vaccine, protecting mice and ferrets from broad viral challenges. This study therefore allowed the identification of potential attenuated vaccine candidates.

Further exploring IAV’s capacity to evade immunity, Wu et al. constructed a mutant library for the vRNA segment of Influenza (H1N1) A/WSN/1933 which encodes NS1 [138], given the critical role of NS1 as an antagonist of the innate immune response [137]. The virus was passaged in the presence or absence of type I interferon in A549 cells. Multiple SNPs displayed >2-fold higher interferon sensitivity compared to WT virus. In particular, D92Y of NS1 showed 12-fold higher sensitivity to the interferon treatment. This mutant was demonstrated to disrupt a hydrophobic pocket on NS1, which appeared to be critical for anti-interferon activity. Further research demonstrated a critical role for the domain in inhibition of RIG-I ubiquitination, a requirement for functional interferon signalling [139].

DMS is an ideal technique for discovery of single-residue changes that impact on viral fitness. However, it can be challenging to investigate paired mutants of significance, even if the mutants happen to be present on a single genome within a DMS experiment dataset. Wu et al. combined a high-throughput genetic study with analysis of naturally occurring sequences to identify a compensatory mutation in matrix protein M1 of IAV [140]. M1 acts as an endoskeleton for the virus [141] and aids virus budding [142]. Transfection of HEK 293T cells with Influenza (H1N1) A/WNS/1933 with a mutant M segment, followed by infection of A549 cells allowed identification of deleterious mutants. Thirteen deleterious mutants were validated, and it was found that 3 of these were prevalent in nature, including M1 Q214H. The software CAPS (Coevolution analysis using protein sequences) allows for identification of co-evolving amino acids with a time-dependent analysis of the evolution of pairs of amino acids from a changing sequence [143]. The analysis using CAPS revealed that M1 214 coevolves with residues 121, 207, 209 and 214. The A209T mutation was found to be compensatory for Q214H, revealing a novel pairwise epistatic interaction.

### 7.5. Severe Acute Respiratory Syndrome Coronavirus 2 (SARS-CoV-2)

#### Variant Analysis

The recent devastating outbreak of SARS-CoV-2 prompted the application of DMS to investigation of binding of the viral spike protein to the angiotensin-converting enzyme 2 (ACE2) receptor [144]. Specifically, this study explored the effects of all amino acid mutations of the receptor-binding domain (RBD) of a Wuhan-Hu-1 SARS-CoV-2 isolate on its interaction with angiotensin-converting enzyme 2 (ACE2) to identify potential targets of future antiviral strategies [145]. Two mutant libraries were prepared and expressed in a yeast-surface display platform. RBD mutant expression was measured using a fluorescent tag and sorted by flow cytometry based on expression levels into bins for analysis by deep sequencing. ACE2 binding was then measured by incubation of cells with fluorescently labelled ACE2, which was separated by flow cytometry into bins for further deep sequencing. Analysis revealed considerable mutational tolerance of RBD for ACE2 binding and normal expression levels. Mutants enhancing affinity for ACE2 were found at RBD sites Q493, Q498 and N501. After publication, the SARS-CoV-2 variant VOC-202012/01, containing an N501Y mutant, spread rapidly throughout the UK and South Africa, indicating the real-world applicability of DMS [146]. In a follow up study, these RBD mutant libraries were tested for escape from 10 human monoclonal antibodies [147]. Escape mutants were identified using the yeast display platform model, with FACS determining ACE2 binding levels. Most escape mutants appeared at the receptor-binding motif, where ACE2/RBD contact residues are present. Negative-stain electron microscopy was used to determine antigenic regions of the RBD for each antibody. This information allowed prediction of effective antibody cocktails which do not allow enrichment of escape mutants

## 8. Concluding Remarks

DMS is an incredibly powerful tool for rapid and high-throughput mutational analysis of viral proteins and genomes, and the above studies have illustrated the plethora of applications that DMS offers to virology. DMS has allowed the construction of attenuated vaccines, evolutionary studies, identification of functional regions of proteins, prediction of mutants that increase host range and many other applications. As the feasibility of NGS increases due to decreases in the cost of such studies, we will likely have a very detailed view of the impact of point mutations for many viral genes. We hope that the recent development of bioinformatic tools and databases which facilitate visualisation and sharing of gathered DMS data will enable and encourage researchers to utilise existing data for further experimentation, akin to the Protein Data Bank. The DMS-based discovery of a SARS-CoV-2 N501 RBD mutant with enhanced ACE2-binding affinity, prior to its natural appearance as a highly trasmissible variant reinforces the utility of DMS. However, it also illustrates the need for strict biosafety measures in the field, in order to prevent the accidental or intentional release of laboratory-created gain-of-function mutants. It has been suggested that publication of specific gain-of-function mutations should be regulated to ensure they are not recreated and misused, with data available on request. Alternatively, as some have argued, studies which may produce gain-of-function mutations in viruses with pandemic potential should be carefully scrutinised and regulated. For additional reading, we refer to two Editorials for further thoughts on the ethical considerations of gain-of-function experiments [148,149].

## Figures and Tables

**Figure 1 viruses-13-01020-f001:**
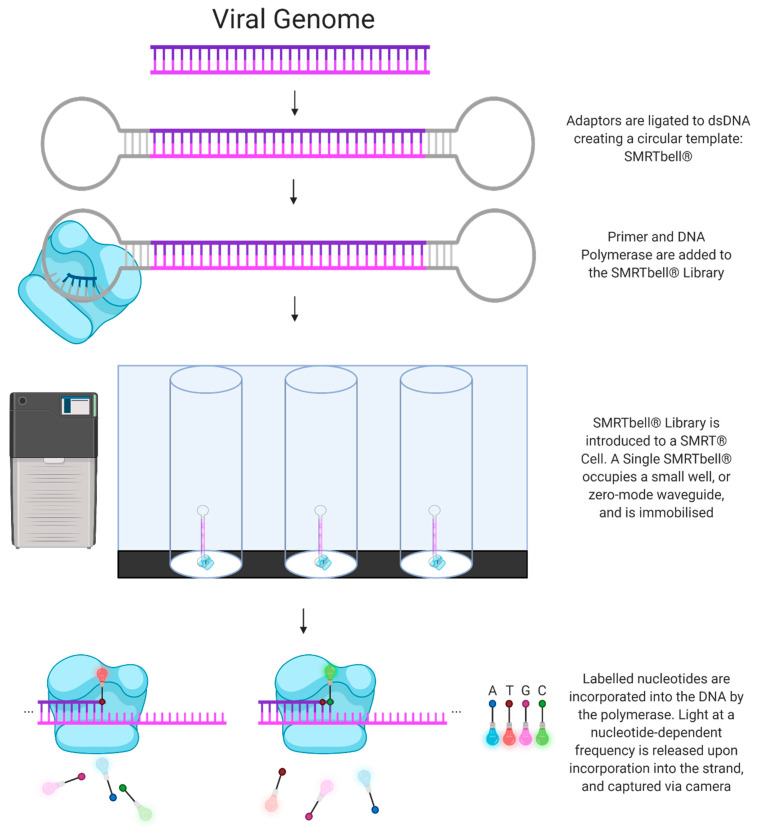
Next-Generation Sequencing: PacBio SMRT Technology.

**Figure 2 viruses-13-01020-f002:**
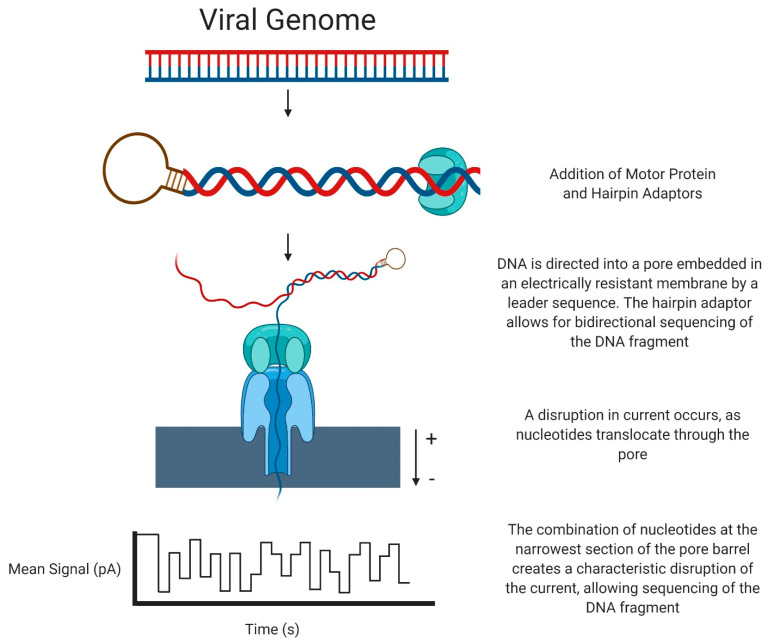
Next-Generation Sequencing: Oxford Nanopore Technology.

**Figure 3 viruses-13-01020-f003:**
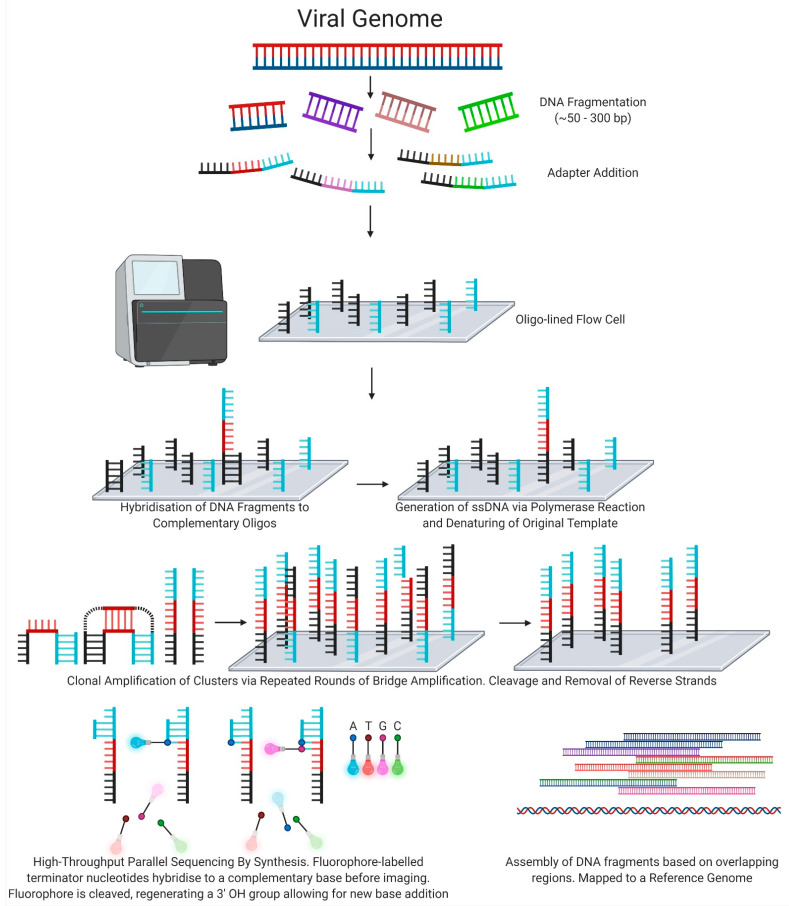
Next-Generation Sequencing: Illumina short-read technology.

**Figure 4 viruses-13-01020-f004:**
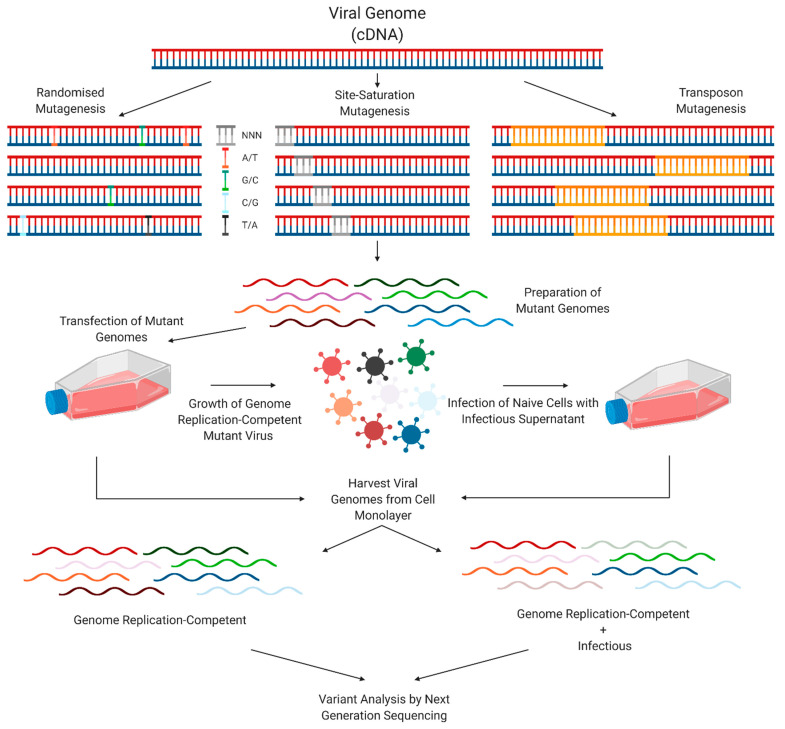
Standard methodology of a DMS experiment.

## Data Availability

Figures are created with BioRender.com.

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
