# Peer review of "Applications of Deep Mutational Scanning in Virology"

_viruses, 2021, doi:10.3390/v13061020_

Round 1

Reviewer 1 Report

In this manuscript, the authors review the uses of deep mutational scanning to explore viral infection, replication, and immune evasion. The present a short description of DMS, viral rescue systems, long read deep sequencing technology, and a long list of examples of viral DMS. This is a comprehensive and well-written review that would be acceptable in its current form.

However, a few tweaks might make it more impactful. The introduction could spend a bit more time explaining the basic principles of DMS and comparing it to prior, less high-throughput mutational screening approaches. The viral rescue and sequencing sections could also use a bit more context. What are important qualities of a rescue system for DMS (i.e. efficiency and capacity to represent complex mixtures)? It does not seem necessary to go into too much detail about rescue system technology, but if T7 and SP6 promoter driven positive stranded virus rescue systems are described, one should also mention that Pol II promoters can also be used.

For the deep sequencing section, the authors could provide more context. For instance, why are long reads are required for DMS. How do these technology compare (i.e. length, depth, error rates, cost)? What is important for DMS?

Some minor editing is also in order, such as removing some double periods and minor formatting issues. 

Reviewer 2 Report

I have read with interest and appreciation the paper entitled “Applications of Deep Mutational Scanning in Virology “ authored by Burton and Eyre. I have found the paper suitable for publication in Viruses  because it describe a current approach in virology and it is relatively comprehensive. I think Viruses readers will benefit from this review.  A major comment is that the paper omits the description of biosafety measures to prevent the spread and potentially ominous release of gain-of-function recombinant viruses.  In the same token, a succinct description of strategies to publish findings that prevent their misuse through reverse genetics would be suitable.

Specific comments

  • 1.1 Line 54, “prophylaxis and control” better describe the idea. 
  • 1.2 Line 71 and 72. For most viruses a vaccine is not available, thus  delete the sentence starting  “For most Flaviviridae…” furthermore effective vaccines against YFV and JEV exist , are very effective and despite error-prone polymerases no neutralization-escape mutants have developed in the population. 
  • 2.1 Line 95 Please add a hammerhead or hepatitis delta ribozyme and/or a T7 terminator to the key ingredients to generate an infective plasmid for reverse genetics systems. Those are essential to generate RNAs with precise ends.
  • Figure 1 needs to have the same quality/readability as figure 2, please scale accordingly.
  • Paragraph in lines 179 to 181 needs to be moved to 2.2 section.
  • Experiments detailed in lines 335  to 350 need to disclose that domain III in Zika virus Envelope corresponds to the cell-receptor interacting site while the elegance of the approach illustrates that regions far from the interacting site also affect the susceptibility to viral neutralization. 
  • The sentence comprised in lines 387 to 292 is very large and difficult to read, please edit it. 
  • Line 392, which termini amino or carboxyl? Which proteins? All DENV structural and non-structural?
  • Line 394, in which cell lines does this statement applies to? Viral fitness is cell-dependent.
  • Line 594 please edit so that it reads: “allowed the identification of potential attenuated vaccine candidates’.  An effective attenuated vaccine is more than selective viral fitness.
  • The manuscript has lots of double periods..

Reviewer 3 Report

The review by Burton and Eyre attempts to serve as a survey of DMS approaches to studying viruses. The authors provide a reasonably comprehensive set of references on the topic, but their descriptions of the underlying work are frequently incomplete, to the point of being difficult to follow even for an expert in the field. That leads me to wondering who this review is for? The authors do not accurately describe the sequencing technologies nor the pro’s and con’s of their use, do not comprehensively describe the challenges in the field, allude to the utility of these approaches but do not compare and contrast with other methodologies, and do not provide a blueprint for others to follow who are interested in pursuing DMS for their own purposes. The review would be well-served with some agenda to inform, which I do not currently find. Given the title, perhaps the authors would be best served by specifically describing advances attributable to DMS, describing the advantages over other, more traditional methods. The authors also fail to describe two incredibly significant contributions of DMS to virology; the study of isolated protein functions in orthogonal systems to understand basal functional constraints, and the use of DMS technology towards serology. I think if the authors focus they could have a great review, but as-written I do not feel like this review will have much impact on informing the field.

Some specific criticism.

  • Line 40: Completely ignores DMS used on individual proteins for protein engineering/evolutionary modeling. While not the topic at hand, to undermine the collective DMS work by claiming the major focus has been viral genomes invites significant criticism.
  • Lin 53: Antigenic drift and shift are not methods of accumulating mutation. They are names attached to mechanisms of change, nothing more. This might just be a word-choice/grammatical error, but I would caution the authors that ascribing intent onto evolutionary mechanisms is not acceptable.
  • Line 59. There are four “species” of influenza virus, A, B, C, and D. Reassortment does not make a new species. This nomenclature is incorrect.
  • Line 66. How? How does DMS inform novel subtype emergence? I mean, there are elements where it could be informative (zoonotic barriers, comparative serology) but just dropping this statement as self-evident when it really isn’t…
  • Line 90. I think it can be taken as a given that a plasmid has an ORI, given that if it doesn’t have an ORI it isn’t a plasmid by definition.
  • Lines 130-135. Why the focus on long-read technology? If the authors really want to go this route maybe actually describe the benefits/drawbacks of such an approach? When it is appropriate versus inappropriate? Many DMS studies used short-read Illumina quite successfully, with the drawback more an inability to study epistasis due to lack of linkage. Also, commonly used platforms? Illumina short read is by far the most common. MinIon and PacBio have been used (and for good reasons) but are definitely not “standard”. As for synthetic illumina long read…I do not believe it has ever been used for DMS? I do not know if synthetic long-read mapping would even be ok for DMS, as it generally requires quite large and diverse DNA input like a human genome—its original iteration was counter-indicated for viral genomes and I am unaware if that fundamental problem has been addressed. If so, the authors could inform us, otherwise I believe they have misconstrued short-read and synthetic long-read technologies. Critically, multiple DNA molecules co-occupy a micelle, but with high diversity fragments derived from disparate molecules can be reassembled. I really think the authors meant to discuss illumina short-read sequencing, but this degree of error is worrying.
  • Figure 3. If indeed they meant to do long-read and I have missed papers using such an approach, the authors give an incorrect description of synthetic long read. So first of all, most Illumina sequencing is short read, which is somewhat described here save there is no absolute size requirement at 350bp but rather a range depending on sequencing considerations (read-length, cluster efficiency etc.). The synthetic long-read requires barcoding within the micelle to reconstruct fragments derived from the same molecule. They have nothing to do with reference mapping, which is quite doable without a synthetic long-read approach. It is abundantly clear here that the authors do not have an understanding of this sequencing approach, so why they wanted to include this, incorrect, description in this review escapes me.
  • Line 200. Multiple mutations are only resolvable by long-read sequencing strategies. Short-read lose linkage and thus cannot be used for epistasis. Again, I think the authors did not know what illumina sequencing is? If I instead presume that is what they mean it is correct.
  • Line 208-209, explain NDT and DBK degeneracy. I wouldn’t assume your readers understand the degenerate nucleotide codes.
  • Lines 235-237, this sentence is a bit vague. Why not discuss insertional toleration more explicitly?
  • Line 435. You discuss tolerance but never define it. If the reader knows what this is, they are probably familiar with DMS, so please define.

Round 2

Reviewer 3 Report

I thank the authors for seriously considering my suggestions, and agree this is a much-improved manuscript. I have only two small textual changes that I think would improve clarity.

lines 240-245. Another way in which this has been addressed is by "subassembly", wherein a specific barcode is associated with mutatations and so epistasis can still be explored by short-read technologies.

Lines 318-320. Wording awkward. The way it is written it sounds like growing more bacteria will increase diversity. I understand the authors mean that the diversity is constrained to a number of uniquely transformed plasmids, but as-written it does not say that.

No remaining major comments. Looking forward to seeing the final product.
